# Integration of Extended Reality and a High-Fidelity Simulator in Team-Based Simulations for Emergency Scenarios

**Youngho Lee** [1], **Sun-Kyung Kim** [2,3,4,*], **Hyoseok Yoon** [5,*], **Jongmyung Choi** [1], **Hyesun Kim** [6] **and Younghye Go** [7]

1   Department of Computer Engineering, Mokpo National University, Muan-gun 58554, Korea; youngho@ce.mokpo.ac.kr (Y.L.); jmchoi@mokpo.ac.kr (J.C.)
2   Department of Nursing, Mokpo National University, Muan-gun 58554, Korea
3   Department of Biomedicine, Health & Life Convergence Sciences, BK21 Four, Mokpo National University, Muan-gun 58554, Korea
4   Biomedical and Healthcare Research Institute, Mokpo National University, Muan-gun 58554, Korea
5   Division of Computer Engineering, Hanshin University, Osan-si 18101, Korea
6   Department of Nursing, Hyejeon University, Hongseong 32244, Korea; twins815@hanmail.net
7   Biomedical and Healthcare Research Institute, Mokpo National University, Muan-gun 58554, Korea; annuhbung@naver.com
*   Correspondence: skkim@mokpo.ac.kr (S.-K.K.); hyoon@hs.ac.kr (H.Y.)

**Abstract:** Wearable devices such as smart glasses are considered promising assistive tools for information exchange in healthcare settings. We aimed to evaluate the usability and feasibility of smart glasses for team-based simulations constructed using a high-fidelity simulator. Two scenarios of patients with arrhythmia were developed to establish a procedure for interprofessional interactions via smart glasses using 15-h simulation training. Three to four participants formed a team and played the roles of remote supporter or bed-side trainee with smart glasses. Usability, attitudes towards the interprofessional health care team and learning satisfaction were assessed. Using a 5-point Likert scale, from 1 (strongly disagree) to 5 (strongly agree), 31 participants reported that the smart glasses were easy to use ($3.61 \pm 0.95$), that they felt confident during use ($3.90 \pm 0.87$), and that that responded positively to long-term use ($3.26 \pm 0.89$) and low levels of physical discomfort ($1.96 \pm 1.06$). The learning satisfaction was high ($4.65 \pm 0.55$), and most (84%) participants found the experience favorable. Key challenges included an unstable internet connection, poor resolution and display, and physical discomfort while using the smart glasses with accessories. We determined the feasibility and acceptability of smart glasses for interprofessional interactions within a team-based simulation. Participants responded favorably toward a smart glass-based simulation learning environment that would be applicable in clinical settings.

**Keywords:** arrythmia; smart glasses; extended reality; team-based simulation; nursing; undergraduate





## 1. Introduction

Arrhythmia is a critical condition, and nurses should be prepared for emergencies relating to it [1]. In an emergency, healthcare providers must be sufficiently skilled to establish and share common knowledge in order to make informed decisions; however, multiple health professionals may have different priorities, thus training for optimal interprofessional interactions would ensure the best patient outcome by providing fast but accurate care in a cooperative manner [2]. Knowing the importance of such issues, organizations such as the American Heart Association, Canadian Cardiovascular Society, and European Society of Cardiology [3,4] have provided guidelines for optimal care. These guidelines are best used when patient data are well collected and analyzed comprehensively, enabling care teams to arrive at an optimal decision [5].

Findings from meta-analyses indicate that simulation-based learning is far more effective than traditional theory-based learning, improving competency and problem-solving ability [3,6]. Currently, high-fidelity simulators have been widely used in nursing schools to focus on reproducing symptoms of illnesses so that students can learn how to care for individual patients [3]. A high-fidelity simulator is a computerized whole-body mannequin that can be programmed to provide dynamic physiological feedback and realistic responses [7]. High fidelity alone, however, does not sufficiently build a clinical situation where diverse health professionals are able to collaborate in a large range of hospital environments.

"Team-based simulation (TBS)" has been introduced to foster collaborative teamwork and mimic possible challenges that may be encountered while working together with colleagues [7–9]. TBS is aimed at establishing a collaborative learning environment in which students become proficient and actively participate in learning team care for complex health problems [9]. Although the benefits of TB—which include improvements in learners' knowledge, problem-solving skills, and critical thinking skills [8]—have been well recognized, current nursing simulation-based education is subject to potential limitations and questions concerning the full engagement of all students involved. Due to high teacher–student ratios and spatial deficits, four to five students generally make up a group, a number that is unlikely to allow the necessary decision-making and interactions required between members [10].

*Related Work*

With the advent of new technology, methodologies for interprofessional interactions have undergone a rapid transformation. With the outbreak of the coronavirus disease in 2019, interest in virtual communication has never been higher [11]. Wearable devices such as smart glasses are considered to be promising, affordable, and easy-to-setup assistive tool for exchanging information in healthcare settings, owing to their intuitive designs, unobtrusiveness, minimal training requirement, ability to facilitate telemedical consultations [12], and general-purpose in-view recognition [13]. Smart glasses display interactive images on the visual field of users by overlaying visual information without significant hinderance of natural vision [14,15]. For this reason, it is time to build a learning environment that incorporates cutting edge technology into education, and to enable students to quickly adapt to a rapidly changing clinical environment.

The recent commercialization of wearable devices has introduced the use of smart glasses in healthcare settings, including nursing education [16–19], clinical simulations [20–22], and surgical telementoring [23–25]. Google Glass (GG) and Microsoft HoloLens are two commercial smart glasses that are actively being studied for medical and industrial applications [26]. Previous studies have demonstrated the use of smart glasses as assistive devices for self-training and informative, "always-on" views. However, most smart glass applications focus on single users, and there is room for improvement in situations involving multiple users or team-based communication. Surgical telementoring involves at least two users, known as the mentor and the mentee, who remotely provide expert guidance to trainees or students. Recent studies [23–25] have used the Microsoft HoloLens to directly establish a communication channel between the mentor and mentee, which can be considered a simplified TBS. A recent review [27] identified areas where mixed reality is used in nursing education. With mixed reality technology, information can be delivered via visual, auditory, and haptic senses, engaging users in simulations. Compared to previous studies, this work explores the use of the latest Glass Enterprise Edition 2 for use in interprofessional interactions (i.e., four individuals per team) in a TBS context, considering the actual use of extended reality in clinical practice. Table 1 compares previous studies in terms of smart glasses, TBS, and targeted domains.

**Table 1.** Summarization and comparison of previous studies.

| Reference | Smart Glass | TBS | # of Persons per Team | Target Domain |
|---|---|---|---|---|
| Chaballout et al. (2016) [21] | Google Glass | No | N/A | Clinical Simulation |
| Iqbal et al. (2016) [22] | Google Glass | No | N/A | Clinical Simulation |
| Vaughn et al. (2016) [20] | Google Glass | No | N/A | Clinical Simulation |
| Kopetz et al. (2019) [17] | Google Glass | No | N/A | Nursing Education |
| Mitsuno et al. (2019) [23] | MS HoloLens | Yes | 2 | Surgical Telementoring |
| Klinker et al. (2020) [28] | MS HoloLens | No | N/A | Healthcare |
| Liu et al. (2020) [24] | MS HoloLens | Yes | 2 | Surgical Telementoring |
| Ingrassia et al. (2020) [29] | MS HoloLens | No | N/A | Medical Training |
| Szajna et al. (2020) [26] | MS HoloLens | No | N/A | Manual Wiring Production Process |
| Kim et al. (2021) [16] | Vuzix Blade | No | N/A | Nursing Education |
| Yoon et al. (2021) [18] | Glass Enterprise Edition 2 | No | N/A | Nursing Education |
| Frederick and Gelderen (2021) [19] | Google Glass | No | N/A | Nursing Education |
| Gasques et al. (2021) [25] | MS HoloLens | Yes | 2 | Surgical Telementoring |
| This work (2021) | Glass Enterprise Edition 2 | Yes | 4 | Clinical Practice |

Given the increased accessibility and improved affordability, there is empirical evidence suggesting the use of smart glasses could result in promising results by enhancing the learning process. The educational application of smart glasses in simulation training has shown strong potential in the acquisition of clinical skills and knowledge [20,30,31]. Provision of supportive information via smart glasses allows confident decision-making during simulations. Furthermore, the ability to conduct real-time discussions and consultations would increase students' willingness to participate in clinical situations and would reduce the burden of uncertainty. This study aims to examine whether adaptation to smart glasses would produce beneficial effects on learning outcomes with TBS by improving team dynamics and learning engagement.

## 2. Methods

### 2.1. Simluation Intervention

The smart glass-based TBS consisted of four parts: (1) lecture, (2) students developing an algorithm for patient care, (3) team-based training, and (4) evaluation. Participants participated in a lecture for electrocardiogram (EKG) analysis, medication, and nursing care for patients with arrhythmia (Table 2). Two scenarios of arrhythmia were introduced, and students were tasked with building a step-by-step algorithm including a decision-making process. Two scenarios of patients with arrhythmia (A-fib and paroxysmal supraventricular tachycardia) who were admitted for emergency care were used. The framework of the 15-h simulation was developed based on a previous study [32] that included seven nursing skills and four to five incidents where students were required to perform nursing practice and make clinical decisions. Since the majority of students had little previous experience using smart glasses, instructions regarding this cooperation system were provided prior to simulation education.

Given the learning objective, which was to improve the ability to practice safe and quality care, the current simulation was designed to administer nursing interventions directly to a high-fidelity simulator; optimal clinical decisions for patients with arrythmia were taken as a team. In each scenario, there were four students in one group: two students played the role of remote supporters, who were in charge of sharing information with two bedside workers for optimal decision-making. Students were provided opportunities for repetitive training using augmented reality (AR), with the option to fix their algorithm. Four teams participated in the experience at a time; participants in three teams observed the performance of the team performing the simulation to correct their algorithm or practice communication and nursing skills for their simulation.

**Table 2.** The simulation framework for team-based simulation based on a scenario of patients with arrhythmia.

| | Simulation Framework | |
|---|---|---|
| Teamwork | Bedside Trainee | |
| | - | Sharing patients' problems and complaints in a clear and effective manner |
| | Remote navigator | |
| | - | Clear allocation of roles and facilitation of cooperation |
| Skill performance | Nursing intervention | |
| | - | Monitoring, EKG, defibrator, oxygen therapy, intravenous cannula, medication, patient education |
| | - | Checking physician order |
| Decision making | 1. | Knowledge and recognition of the condition (determining the types of arrhythmias) |
| | 2. | Assessment of the severity and urgency of the situation (patients who require urgent or emergent care) |
| | 3. | Effectiveness of medication |
| | 4. | Patient's condition for additional treatment |
| | 5. | Patient's condition for urgent cardioversion |
| Knowledge | Knowledge of arrhythmia | |
| | - | Signs and symptoms of each arrhythmia |
| | - | Knowledge of medication for arrhythmia |
| | - | Interpretation of electrocardiogram |
| | History of patient | |
| | - | Current medication, underlying disease, etc. |

Participants exchanged their roles and completed one of two scenarios as remote supporters. Each scenario comprised five tasks, and a team was required to make five decisions. Depending on the capacity and teamwork, an individual scenario was run for 20–25 min. The duration was shortened as the participants repeated the practice.

Using checklists, each participant's performance was assessed in terms of accuracy and adequacy of skill performance and decision-making process. The participants were provided a questionnaire regarding their experience with smart glass and overall simulation on completion of the TBS.

### 2.2. Development

Our system was composed of a trainee-side wearable system, supporter-side desktop system, and a network server (Figure 1). The trainee-side wearable system comprised a GG, Bluetooth earphones, and a small mirror attached to the GG. The GG had a front camera, Bluetooth audio, and a touchpad. The supporter-side desktop system consisted of a desktop computer and headset, video conferencing application program to monitor the trainee, and image transmission program for sharing image files.

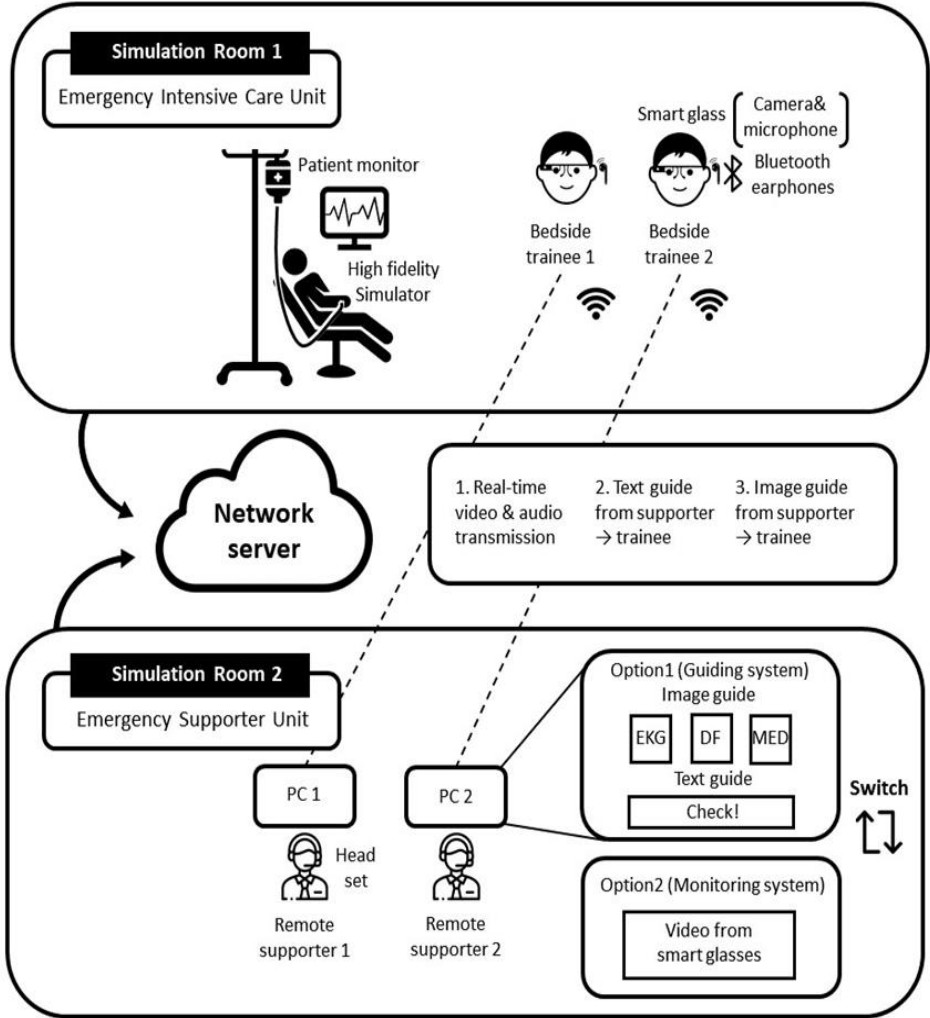

**Figure 1.** The flow of the team-based simulation using smart glasses and a high-fidelity simulator.

The trainee-side wearable system transmitted real-time video and audio captured from the trainee's field of view through the GG, and received audio and image transmitted from the supporter-side system. We used Google Glass Enterprise Edition 2 (Glass EE2; Android Oreo 8.1). The supporter-side desktop system received the trainee's video and enabled voice communication. We also developed software that could transmit images and text required for training by accessing the web RTC server using Google Chrome.

Our system was installed in two simulation rooms: the emergency intensive care and emergency support units (Figure 2). The two trainees in the emergency intensive care unit wore a trainee-side wearable system and a high-fidelity patient simulator. The two remote supporters in the emergency support unit sat at the supporter-side desktop system. The supporters were able to use two desktop application programs: the image guide and the monitoring system. The image guide delivered selected images to the corresponding trainees and transmitted real-time video from the smart glasses to the supporter monitor.

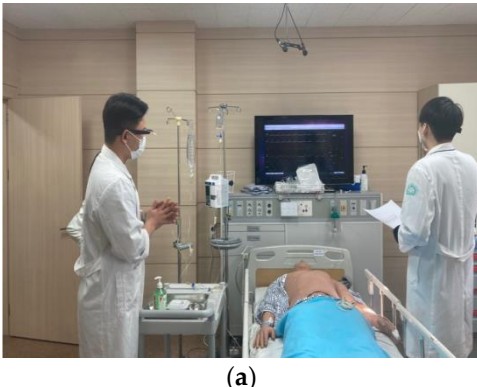 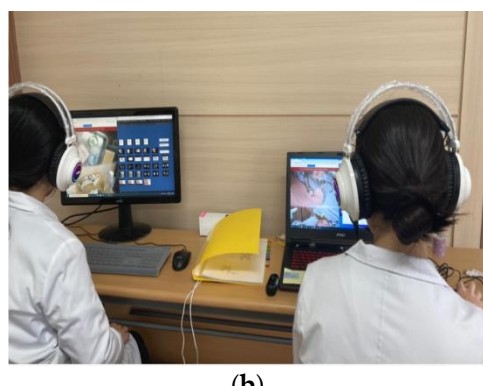

(**a**) (**b**)

**Figure 2.** Room setup for emergency unit simulation: (**a**) in the emergency intensive care unit; (**b**) in the emergency support unit.

The system supported one-way video communication and two-way audio communication, along with image and text message transmission. The video captured by the camera mounted on the trainee's smart glass was transmitted to the supporter's desktop through the webRTC server. While the supporter was able to attach a camera and transmit video, it was not transmitted to the trainee because the supporter's appearance was not relevant to the simulation. The voice signal was acquired from the microphones in the trainee's GlassEE2 and the headset worn by the supporter. Both voices were audible to each speaker. Trainees used Bluetooth earphones because several people were in the same room as them, while the supporters used headsets. The supporter sent images and text messages to help the trainee, which appeared on the display of the GlassEE2.

We developed a network-based data transmission program that transmitted images and text messages from the supporter system to the trainee system. It was developed using Unity3D (ver. 2018.3.14f1) for visualization on the desktop side, and a User Datagram Protocol socket was used for fast multi-party data transmission. If the supporter clicked on the image required by the trainee, the image was delivered to the trainee's GlassEE2. The program also had a text box where the supporter could write a message to the trainee; when the supporter wrote a text message, it immediately appeared on the trainee's GG and disappeared after a few seconds. The following procedure was set out for trainees and supporters participating in the experiment:

Step 1: Preparing for connection

- The trainee wears the GlasssEE2 and Bluetooth earphone and the supporter wears the headset while sitting at the desktop.

Step 2: Connection

- A trainee powers on the GlassEE2 and earphones.
- The trainee runs our application in the GlassEE2.
- The application automatically connects to the webRTC server for video conferencing.
- The trainee informs the supporter of the room number.
- A supporter opens the Chrome browser and inputs the room number into the webRTC server.
- Video conferencing connection is established.
- The supporter runs the image and text message transmission application, and it establishes a connection to the GlassEE2.

Step 3: On study

- The trainee performs the task. See the video, images, and text messages from the trainee.

- The supporter monitors the video and audio from the trainee and talks to the trainee about the task. If the trainee requires further explanation, the supporter sends images or text messages to help the trainee.

### 2.3. High-Fidelity Simulator

The METIman [33] is a life-like human simulator that physically represents patients. With clinical features including breathing, pulse, heart and lung sounds, various scenarios of adults with heart disease can be simulated. Not only does the simulator show physical symptoms, but the monitor connected to the simulator displays associated signs (e.g., vital signs).

### 2.4. Sample

The study population included nursing students who attended a baccalaureate nursing program at a university located in J district, Korea. Thirty-two participants were recruited. After excluding one incomplete survey, data from 31 participants were used for statistical analysis. The purpose of this study was explained to all study participants and written informed consent was obtained.

### 2.5. Instrument

#### 2.5.1. Usability

The instrument developed by Ingrassia et al. [29] was used to evaluate feasibility and acceptability of the system. We revised the questionnaire according to the design and purpose of this study. The survey consisted of 54 questions spanning six categories: user input (12 items), system output (6 items), system usability (17 items), fidelity (8 items), immersivity (4 items), and likability (7 items). Using a 5-point Likert scale, the study participants provided ratings from 1 (strongly disagree) to 5 (strongly agree). There were four additional questions in which study participants rated their overall level of satisfaction from 1 (very dissatisfied) to 4 (very satisfied) regarding the smart glasses device, ease of use, system output, and smart glass-based simulation.

#### 2.5.2. Attitudes towards the Interprofessional Health Care Team

The Attitudes Towards Interprofessional Health Care Teams Scale developed by Heinemann et al. [34] and modified by Curran et al. [35] was used to evaluate the effectiveness of the educational program. This scale consists of 14 items that include quality of care, time constraints, and teamwork among health professionals. This scale uses a 5-point Likert scale with values ranging from "strongly disagree" (1) to "strongly agree" (5).

#### 2.5.3. Learning Satisfaction

Satisfaction with the education program was measured using the questionnaire used by Ji and Chung [36]. A 5-point Likert scales was used, ranging from "not at all satisfied" (1) to "very satisfied" (5). A higher score indicated a higher level of satisfaction with the education program.

#### 2.5.4. Essay Questionnaire

An essay questionnaire was used that asked the following seven qualitative questions to obtain in-depth information about user perceptions of the smart glass-based simulation education:

1. "How did you find the smart glass-based emergency simulation in general?"
2. "Was this program easy to use? What are the points that you thought needed further improvement?"
3. "Were there any difficulties or constraints when operating the system?"
4. "Do you think this smart glass-based simulation education would be useful for your future practice?"
5. "Do you think the smart glass would be useful in the clinical environment?"

6. "What are the components that require additional technical efforts for active application and continuous use of these smart glasses?"
7. "Please add any other comment on this program."

*2.6. Statistical Analysis*

Statistical analysis was performed using the IBM SPSS Statistics version 25.0 (IBM Corp, Armonk, NY, USA) [37]. For data analysis, frequency, percentage, mean, and standard deviation were calculated.

### 3. Results

*3.1. Quantitative Response*

The mean age of the study participants was 23.9, and 45.1% of participants did not have previous experience with AR technology (Table 3). The participants reported good control over the system overall (mean 3.77, SD 0.67). Some participants expressed a lack of controllability (22.6%) and a preference for alternative interaction methods (38.7%). A majority of the participants (87.1%) reported that the display of the system was appropriate. In terms of the content provided by the smart glass screen, 58.1% of participants experienced some degree of limitation in seamless performance of tasks (Figure 3).

**Table 3.** General characteristics of study participants ($N = 31$).

| | | M ± SD or n (%) |
|---|---|---|
| Age (years) | | 23.9 ± 1.24 |
| Gender | Female | 23 (74.2) |
| | Male | 8 (25.8) |
| Grade | Poor | 5 (16.1) |
| | Fair | 20 (64.5) |
| | Good | 6 (19.4) |
| Major satisfaction | Poor | 0 (0) |
| | Fair | 8 (25.8) |
| | Good | 23 (74.2) |
| Previous use of Augmented reality | Never used | 14 (45.1) |
| | A few times | 11 (35.5) |
| | Once a month | 4 (12.9) |
| | Once a week | 2 (6.5) |
| | Every day | 0 (0) |
| Previous use of smart glasses | Never used | 20 (64.5) |
| | A few times | 11 (35.5) |
| | Once a month | 0 (0) |
| | Once a week | 0 (0) |
| | Every day | 0 (0) |

Participants reported that a certain degree of cognitive load was required to operate the system (mean 2.19, SD 1.14), with a moderate requirement for prior learning for proper use of the program. Most participants felt confident about using the program (mean 3.90, SD 0.87), but a need for technical support was high (mean 3.90, SD 0.79). In terms of the GG, the perceived physical effort for using the device was low (mean 1.96, SD 1.06) with little fatigue on the head (mean 1.55, SD 0.72) or arms (mean 1.42, SD 0.72). Participants were more critical of the eye fatigue caused by the display (mean 2.19, SD 1.17) (Figure 4).

| User Input | Mean±SD | Strong Disagree | Disagree | Neither | Agree | Strong Agree |
|---|---|---|---|---|---|---|
| I had the right level of control over what I wanted to do | 3.77±0.67 | 0 | 6.5 | 16.1 | 71 | 6.5 |
| The effect of my interaction was easy to predict | 3.71±0.74 | 0 | 6.5 | 25.8 | 58.1 | 9.7 |
| Accurate pointing with gaze was easy to achieve | 2.65±1.23 | 16.1 | 38.7 | 19.4 | 16.1 | 9.7 |
| Interaction with the system was fast enough | 3.29±1.22 | 6.5 | 22.6 | 25.8 | 25.8 | 19.4 |
| I found the input modalities ideal for the task to be performed | 3.81±0.70 | 3.2 | 25.8 | 58.1 | 12.9 | 0 |
| The system did not behave as I expected | 2.45±1.12 | 19.4 | 41.9 | 16.1 | 19.4 | 3.2 |
| I could not always achieve what I wanted the system to do | 2.00±0.86 | 29 | 48.4 | 16.1 | 6.5 | 0 |
| I kept making mistakes interacting with the system | 2.42±1.18 | 29 | 22.6 | 29 | 16.1 | 3.2 |
| Hand interactions were difficult to perform | 1.81±1.05 | 48.4 | 35.5 | 6.5 | 6.5 | 3.2 |
| The system responded too slowly during interaction | 2.03±0.87 | 29 | 45.2 | 19.4 | 6.5 | 0 |
| I would have preferred alternative interaction methods | 3.00±1.10 | 12.9 | 16.1 | 32.3 | 35.5 | 3.2 |
| I found the touch pad was too sensitive | 2.48±1.09 | 22.6 | 29 | 25.8 | 22.6 | 0 |
| **System output** | | **Strong Disagree** | **Disagree** | **Neither** | **Agree** | **Strong Agree** |
| I found the display appropriate for the tasks | 3.87±1.05 | 3.2 | 9.7 | 12.9 | 45.2 | 29 |
| I thought the words and symbols on screen were easy to read | 3.68±1.24 | 9.7 | 9.7 | 9.7 | 45.2 | 25.8 |
| I felt the display was too limited to see the contents and carry out the task | 3.39±1.17 | 9.7 | 12.9 | 19.4 | 45.2 | 12.9 |
| Screen was not legible because of light, reflection or glare | 2.48±1.20 | 25.8 | 29 | 19.4 | 22.6 | 3.2 |
| I felt that the display was flickering too much and graphics were too unstable | 1.94±0.89 | 35.5 | 41.9 | 16.1 | 6.5 | 0 |
| The quality of the display affected my performance | 3.00±1.18 | 12.9 | 22.6 | 22.6 | 35.5 | 6.5 |

**Figure 3.** User input and system output (*N* = 31).

| System Usability Scale (SUS) | Mean±SD | Strong Disagree | Disagree | Neither | Agree | Strong Agree |
|---|---|---|---|---|---|---|
| I would use this program frequently | 3.77±0.67 | 0 | 6.5 | 16.1 | 61.3 | 16.1 |
| The program was easy to use | 2.65±1.23 | 3.2 | 9.7 | 22.6 | 51.6 | 12.9 |
| Various functions were well integrated | 3.81±0.70 | 0 | 9.7 | 19.4 | 54.8 | 16.1 |
| The most people would find this program easy to use | 2.00±0.86 | 3.2 | 3.2 | 9.7 | 51.6 | 32.3 |
| I felt confident using this program | 1.81±1.05 | 0 | 6.5 | 22.6 | 45.2 | 25.8 |
| The program was unnecessarily complex | 3.71±0.74 | 32.3 | 35.5 | 16.1 | 12.9 | 3.2 |
| I needed technical support | 3.29±1.22 | 0 | 3.2 | 25.8 | 48.4 | 22.6 |
| There were too much inconsistencies | 2.45±1.12 | 12.9 | 9.7 | 25.8 | 35.5 | 16.1 |
| The program was very cumbersome to use | 2.42±1.18 | 35.5 | 38.7 | 9.7 | 16.1 | 0 |
| I think I need to learn numerous things before using this program | 2.03±0.87 | 6.5 | 9.7 | 19.4 | 32.3 | 32.3 |
| **Google Glass 2nd Edition** | **Mean±SD** | **Strong Disagree** | **Disagree** | **Neither** | **Agree** | **Strong Agree** |
| I think I can use the device for long time | 3.26±0.89 | 0 | 22.6 | 35.5 | 35.5 | 6.5 |
| The Google Glass is too big and heavy | 1.77±0.88 | 48.4 | 29 | 19.4 | 3.2 | 0 |
| The device needs substantial mental efforts (concentration) to operate | 2.65±1.43 | 32.3 | 16.1 | 16.1 | 25.8 | 9.7 |
| The device needs great physical efforts to operate | 1.94±1.06 | 45.2 | 29 | 12.9 | 12.9 | 0 |
| There was high fatigue in arm, hands and fingers | 1.42±0.72 | 67.7 | 25.8 | 3.2 | 3.2 | 0 |
| There was very high fatigue in eyes | 2.19±1.17 | 38.7 | 22.6 | 19.4 | 19.4 | 0 |
| There was very high fatigue in head | 1.55±0.72 | 58.1 | 29 | 12.9 | 0 | 0 |

**Figure 4.** System usability of the smart glass-based TBS program (*N* = 31).

The study participants reported good usability of the information provided via the smart glass (mean 4.13, SD 0.67), reporting that it was useful for problem solving within the simulation scenarios. Four participants (13%) reported a lack of realism in the smart glass-based TBS. Apart from one study (3.2%), all participants expressed good engagement in the simulation (Table 4).

**Table 4.** Fidelity, immersion, and presence of smart glass-based TBS (N = 31).

| | Mean ± SD (1–5) | Strongly Disagree n (%) | Disagree n (%) | Neither n (%) | Agree n (%) | Strongly Agree n (%) |
|---|---|---|---|---|---|---|
| Fidelity of simulation | | | | | | |
| The tasks within the simulation were too simplified | 2.00 ± 1.00 | 11 (35.5) | 12 (38.7) | 6 (19.4) | 1 (3.2) | 1 (3.2) |
| The simulation behaved in a very unusual manner | 2.00 ± 1.21 | 14 (45.2) | 9 (29.0) | 4 (12.9) | 2 (6.5) | 2 (6.5) |
| The content provided via smart glass was not helpful | 2.03 ± 0.98 | 11 (35.5) | 11 (35.5) | 6 (19.4) | 3 (9.7) | |
| The information provided via smart glass was well used within the simulation | 4.13 ± 0.67 | | 5 (16.1) | 17 (54.8) | 9 (29.0) | |
| The information provided via the smart glass was useful for problem solving | 4.26 ± 0.68 | | 4 (12.9) | 15 (48.4) | 12 (38.7) | |
| The quality of graphics was very realistic | 3.48 ± 0.89 | | 4 (12.9) | 12 (38.7) | 11 (35.5) | 4 (12.9) |
| The quality of graphics influenced my performance | 3.26 ± 1.18 | 3 (9.7) | 5 (16.1) | 8 (25.8) | 11 (35.5) | 4 (12.9) |
| I thought that the AR content was properly aligned with reality | 3.74 ± 0.77 | | 1 (3.2) | 11 (35.5) | 14 (45.2) | 5 (16.1) |
| Immersion/Presence | | | | | | |
| The information provided by the smart glass gave me the impression of being somewhere else | 2.19 ± 1.01 | 8 (25.8) | 13 (41.9) | 7 (22.6) | 2 (6.5) | 1 (3.2) |
| I was really engaged in the simulation | 3.77 ± 0.84 | 1 (3.2) | | 9 (29.0) | 16 (51.6) | 5 (16.1) |
| The quality of the graphics reduced the level of immersion/sense of presence | 2.16 ± 1.10 | 9 (29.0) | 14 (45.2) | 3 (9.7) | 4 (12.9) | 1 (3.2) |
| The display field-of-view reduced the level of immersion/sense of presence | 2.29 ± 1.13 | 8 (25.8) | 12 (38.7) | 7 (22.6) | 2 (6.5) | 2 (6.5) |

A majority of the participants reported that they had a pleasant experience (mean 4.65, SD 0.55) and intended to use this type of simulation again (mean 4.77, SD 0.42). Most participants agreed that the smart glass had a high potential for practical benefits (mean 4.52, SD 0.62) and was effective in simulation education (mean 4.52, SD 0.76) (Table 5).

**Table 5.** Likability of the smart glass-based TBS program (*N* = 31).

| | Mean ± SD (1–5) | Strongly Disagree n (%) | Disagree n (%) | Neither n (%) | Agree n (%) | Strongly Agree n (%) |
|---|---|---|---|---|---|---|
| Likability | | | | | | |
| The smart glass-based TBS is pleasant | 4.65 ± 0.55 | | | 1 (3.2) | 9 (29.0) | 21 (67.7) |
| I enjoyed the smart glass-based TBS | 4.39 ± 0.76 | | 1 (3.2) | 2 (6.5) | 12 (38.7) | 16 (51.6) |
| I would participate in this kind of simulation again | 4.77 ± 0.42 | | | | 7 (22.6) | 24 (77.4) |
| I could see practical benefits of this program in health care settings | 4.52 ± 0.62 | | | 2 (6.5) | 11 (35.5) | 18 (58.1) |
| Smart glass-based team simulation is an effective educational strategy | 4.52 ± 0.76 | | 1 (3.2) | 2 (6.5) | 8 (25.8) | 20 (64.5) |
| Smart glass-based team simulation is helpful for my future clinical practice | 4.58 ± 0.67 | | | 3 (9.7) | 7 (22.6) | 21 (67.7) |
| Smart glass-based team simulation meets my educational needs | 4.13 ± 0.99 | | 2 (6.5) | 7 (22.6) | 7 (22.6) | 15 (48.4) |

In terms of the participants' level of satisfaction, the highest scores were obtained on smart glass-based TBS (mean 3.26, SD 0.68) and system outputs were reported as the least satisfying (mean 2.68, SD 0.94) (Table 6).

**Table 6.** User satisfaction of the smart glass-based TBS program (*N* = 31).

| User Satisfaction | Mean ± SD (1–4) | Very Unsatisfied n (%) | Unsatisfied n (%) | Satisfied n (%) | Very Satisfied n (%) |
|---|---|---|---|---|---|
| Smart glass device | 2.81 ± 0.83 | 2 (6.5) | 8 (25.8) | 15 (48.4) | 6 (19.4) |
| Ease of use of smart glass | 2.94 ± 0.77 | 1 (3.2) | 7 (22.6) | 16 (51.6) | 7 (22.6) |
| System output | 2.68 ± 0.94 | 4 (12.9) | 8 (25.8) | 13 (41.9) | 6 (19.4) |
| Smart glass-based TBS | 3.26 ± 0.68 | | 4 (12.9) | 15 (48.4) | 12 (38.7) |

Overall, the attitude towards the interprofessional health care team was scored 4.23 (SD 0.59) and participants scored the "quality of care" subscale relatively higher than the other subscales (Table 7). Table 8 presents the learning satisfaction related to the current smart glass-based TBS. In terms of learning satisfaction, the item "I actively engaged in learning activities" was scored the highest (mean 9.19, SD 1.08) and the item "It is an effective way to achieve the learning goal" was scored the lowest (mean 8.58, SD 1.46).

**Table 7.** Attitude toward the interprofessional health care team (*N* = 31).

| | Category | Mean | SD |
|---|---|---|---|
| Attitudes towards Interprofessional Health Care Team (ATIHCT) | Quality of care | 4.34 | 0.61 |
| | Person centered care | 4.26 | 0.64 |
| | Time constraints | 3.91 | 0.77 |
| | Total score | 4.23 | 0.59 |

**Table 8.** Learning satisfaction of TBS (*N* = 31).

| Item | Mean | SD |
|---|---|---|
| It was interesting | 8.71 | 1.40 |
| It is an effective way to achieve the learning goal | 8.58 | 1.46 |
| It provided a meaningful learning experience | 9.03 | 1.22 |
| It is more effective than theory-based lectures | 8.81 | 1.20 |
| I actively engaged in learning activities | 9.19 | 1.08 |
| I am satisfied with the class overall | 9.10 | 1.38 |
| I would like to participate in other smart glass-based TBS | 9.06 | 1.29 |
| Learning satisfaction total | 62.48 | 7.30 |

*3.2. Qualitative Response*

3.2.1. Overall Experience

About 8 out of 10 (84%) participants responded positively. Users reported that smart glass-based team simulation was new, exciting, and interesting (40%), and that they experienced practical assistance while using the smart glasses during the simulation (44%). Some participants (16%) felt frustrated, reporting that the system needed further improvement.

3.2.2. Perceived Benefits

Thirty participants stated that they expected the smart glasses to improve the accuracy of nursing skill performance using image and voice guidance (96.8%). Seventeen users (54.8%) reported that fast and accurate information exchange using smart glasses could assist interprofessional teamwork. Twenty-six participants (83.9%) reported the system could prevent the occurrence of errors, saying, "The remote supporter found me using the wrong dose of medication." Six participants (19.4%) reported they felt confident using medical devices, saying, "Although I used an EKG device for the first time, image guidance and voice assistance from the remote supporter was really helpful."

*3.3. Recommendation*

The study participants' recommendations for further improvements were largely regarding the narrow field of view provided for remote supporters (46%), lack of smoothness in communication between remote supporters and bedside trainees (45%), and discomfort of the smart glasses, especially for those who wear glasses (13%). When taking the role of remote supporter, participants complained that "It was sometimes hard to figure out the patient's condition. I guess it might be easier with real patients, but it was really difficult when a mannequin was the patient." In terms of communication, most problems were caused by incomplete construction of the network and Bluetooth earphones, and participants reported that "Wearing glass and earphones at the same time seems too much. It is needed to have something that can replace earphones." In addition, users recommended adding a visual status indicator to confirm whether the right information was sent (n = 2, 6.5%), suggesting "It would be nicer to confirm whether the image and text that I clicked and typed were sent. I had to keep asking what the bedside trainees were seeing".

## 4. Discussion

The present study evaluated the usability and feasibility of adopting smart glasses for training of nursing skills and interprofessional interactions in emergency arrhythmia situations. The findings in this study indicate that participants attempted to use various features of the smart glasses to be actively engaged in team working and clinical decision-making to provide the best nursing care. The results of the current study provide evidence of the great potential for smart glasses in clinical settings and interprofessional interactions.

A majority of the students perceived TBS using smart glasses as a new, exciting, and interesting method for learning. This is in line with previous studies that have reported the incorporation of high-tech devices into education as promoting great interest among this digitally native generation [38]. Given the clear roles within the simulation (bedside trainees or remote supporters), chances for participants to take part in team communication were greater, with greater responsibility on the remote supporters.

Among the subscales of attitudes toward interprofessional healthcare teams, the scores for the "quality of care" subscale were greater than those for the others. This could be related to the purpose of smart glass as a means of information sharing and interprofessional interactions that are intended to produce the best patient outcomes. Bedside trainees were connected to the remote supporter constantly and felt less bothered seeking for external help, which users found to be a helpful means of potentially addressing communication breakdown. Most features of the smart glasses are actually meant for a seamless flow of teamwork [39], which be advantageous in emergency care settings where efficient and accurate interprofessional interactions are key to improved patient outcomes.

Generally, the users responded positively regarding the GG itself, expressing good feasibility for long-term use with a low level of physical discomfort experienced. One explanation may be that the weight of the GG EE2 is 46 g, which is relatively lighter than other smart glasses (65–350 g) [40]. However, no previous studies have clearly reported wearability and comfortability of the GG; thus, further comparison studies could contribute to the physical design of wearable devices.

Interestingly, the discomfort experienced by some users was mostly related to the accessories, and only a few participants from previous studies had complaints regarding the physical aspects of the GG (short battery life, heavy weight, etc.) [21,22]. We used Bluetooth earphones and mirrors to compensate for the defects of the smart glasses, which users found to be superfluous. There was consistent frustration expressed by individuals wearing glasses, reporting discomfort due to the double-layering of glasses. Although one of the reasons why the GG was chosen was its lens-free design, it would be better to seek other smart glass solutions.

The findings of this study reveal that the participants were least satisfied with the system output, asking for better resolution and bigger screens. Wearable assistive technologies have different intended uses, environments, and trade-offs [41,42] as those in this study.

In clinical settings, assistive devices should enable the wearer to seamlessly perform the clinical workflows of the main task, benefit from the hands-free potential of the device, and maintain sterile conditions [43]. The small display of the GG limits the wearer's peripheral vision and multitasking capabilities [44] in terms of focus and attention to clinical tasks and workflows. In this study, we observed that the use of unfamiliar devices by inexperienced users within a tense situation could aggravate the difficulties related to the display; thus, it is suggested that sufficient time be provided for users to become accustomed to new devices.

The perceived usefulness of smart glass-based TBS for future clinical practice was high among the study participants. The current system was designed to focus on the exchange of information between remote supporters and bedside trainees, informing them of the high risks of medical errors from unskilled care providers [45]; the participants of this study found the smart glasses to be of great use for interprofessional teamwork. Previous studies have also demonstrated that risk signs in patients were better detected when sharing patient monitoring via smart glasses [43]. In addition, the smart glass was widely used as a supportive system for skill training, and a potential benefit to using these glasses in healthcare settings was identified in previous studies involving CPR [46] and nursing skill training [16]. When provided guidance via the smart glasses, trainees were able to improve their performances and achieve higher competency and success rates compared to those trained without smart glasses.

The participants reported low complexity of the simulation program, but some users complained of low compliance. Delayed audio was reported as an important concern that should be resolved. A previous study identified the requirements of low-latency applications for virtual reality (latency = 1 ms) and tele-surgery (1–10 ms) [47]. Although there were concerns over network issue, this is an area where technology is growing rapidly. Therefore, the development additional smart glass-based interventions for education and clinical use is worthwhile.

Further consideration of user-centered interfaces may be the key to success in future system development. The principles of user-friendly interface design include an understanding the intended users and their needs [48]. Users recommended the additional incorporation of visual status indicators due to the degree of uncertainty related to communication that was experienced within the current system. As this system was designed for use in emergency care settings, it is necessary to minimize the cognitive effort required to verify the actions of users.

*Future Implications*

Overall, the AR system used in this study achieved good usability and likability; users agreed on the potential of this innovative technology in nursing education and clinical practice. Considering the continuous progress in terms of technology, the introduction of new technology at an undergraduate level could foster an innovative and progressive atmosphere in future practice. Quality of care would benefit from the use of cutting edge technologies.

Unsatisfactory responses were generally related to user input, and mainly related to user comprehension of the system and ability to operate the device. In simulation education, no matter how easy it is to use the system, a task to which users are unaccustomed (and new methods of information delivery within an emergency scenario) may result in perceived complexity. At certain times, it is necessary to adapt oneself to new devices; putting individuals in complex situations with new devices may result in them having a negative impression towards the device. Nevertheless, when applying communication systems to clinical practice, the devices should not be distracting to the health professionals. Improvements reflecting user feedback are key to the successful implications of new technology in healthcare.

The user response regarding system output showed that understanding the target user is necessary for the system developer. The lack of integrity of the current system should

be improved. When developing systems for clinical use, misleading users with wrong or inadequate information could lead to negative consequences related to patient safety. A multidisciplinary approach with various stakeholders will help further refine the system and ultimately lead to an optimal system for interprofessional interaction via smart glass in clinical settings.

## 5. Limitations

This study had some limitations. It was difficult to draw clear conclusions while relying on self-reported questionnaires. In addition, the current study design for outcome measures did not sufficiently prove the feasibility of smart glass-based interprofessional interaction within actual clinical settings.

## 6. Conclusions

Smart glasses were introduced as an effective means of communication in the industry. In this study, we presented a TBS using GG to investigate the usability and feasibility of smart glasses for interprofessional interaction. Simulations of patients with arrhythmia were created. Nursing students played the roles of bedside trainees and remote supporters, and used GG to share knowledge and communicate. The findings of this study indicate that the current TBS program may help improve simulation-based education and be useful in clinical settings. Some drawbacks were identified that require further consideration and improvement for future studies.

**Author Contributions:** Conceptualization, Y.L., S.-K.K., H.K. and J.C.; methodology, S.-K.K. and H.Y.; formal analysis, Y.L. and S.-K.K.; investigation, Y.L. and S.-K.K.; writing—original draft preparation, Y.L., S.-K.K., H.K. and J.C.; writing—review and editing, H.Y., S.-K.K., Y.G. and Y.L.; visualization, H.Y., S.-K.K., Y.G. and Y.L. All authors have read and agreed to the published version of the manuscript.

**Funding:** This research was supported by a grant (20012234) of Regional Customized Disaster-Safety R&D Program funded by the Ministry of Interior and Safety (MOIS, Korea). This work was supported by a National Research Foundation of Korea (NRF) grant funded by the Korea government (MSIT) (No. NRF-2018R1D1A1B07048247, No. NRF-2018R1D1A1B07043983).

**Institutional Review Board Statement:** The study was conducted according to the guidelines of the Declaration of Helsinki, and approved by the Institutional Review Board (or Ethics Committee) of Mokpo National University (MNUIRB-201006-SB-011-02/12.Oct.2020).

**Informed Consent Statement:** Written informed consent was obtained from study participants before the commencement of study.

**Data Availability Statement:** The data presented in this study are available on request from the corresponding author.

**Acknowledgments:** We express sincere gratitude to the participants in the experiments.

**Conflicts of Interest:** The authors declare that this work has no competing interest.

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
