# Peer review of "Integration of Extended Reality and a High-Fidelity Simulator in Team-Based Simulations for Emergency Scenarios"

_electronics, doi:10.3390/electronics10172170_

Round 1
Reviewer 1 Report
The paper addresses the hottest topic of using extended reality solutions in team-based simulation for nursing students.
In more detail, the paper discusses a study whose purpose was to evaluate the usability and feasibility of adopting smart glasses for nursing skills and interprofessional interactions in emergency arrhythmia situations.
The main contribution is the execution of a user study that simulates actual working conditions toward using a high-fidelity simulator.
Arguments in favor:
- The topic is exciting; new technology fostered new opportunities in collaborative teamwork that need to be explored in detail.
- The authors clearly present the related works and introduce the realized system composed of a trainee-side and a supporter-side interconnected by a network server.
The system's goal is to promote common knowledge sharing to make informed decisions during emergencies.
Arguments against:
- The paper's main contribution is represented by the user study performed and presented in the Methods section. However, in my opinion, the Methods section should be improved to better exposing the procedures adopted in performing the tasks. Some information is unclear or missing:
- Authors should better introduce the used technical instruments.
Since the questionnaire also evaluates the interaction modalities, these should be presented too to the reader.
- It is not clear to me who answered the questionnaires. Are the questionnaires proposed to both the trainee side and the support side users?
- Table 2 introduces participants by dividing them into two groups: intervention and control. The difference between the two groups is not clear since they are not mentioned anywhere in the paper.
- How long did the subjects practice with the tested systems?
- How long was each task?
- How many times does the tester execute the test?
- How were distributed the scenarios among the tests?
- How much resting time between two consecutive tasks (if any)?
- Moreover, in my opinion, some of the results should also be presented using graph representation; this will improve the paper's readability.
- The real weakness of the study is the absence of a data analysis oriented to discover possible relations among the users' answers. Authors only analytically discuss the results by considering one answer at a time.
- Finally, in the result discussion, it is unusual to see so many citations. This section is intended to discuss the achieved results and bringing out both the critical concepts and the of non-obvious entities.
In conclusion, the paper deals with an interesting topic, but it needs to be improved before it is ready for publication.
Typos:
- raw 48: "7ability".
- raw 141: Instrument subsection (2.5) reports almost the same content of Statistical Analysis subsection (2.6).
Author Response
We are grateful for the positive and constructive comments that originated for the review process.
We have responded to the comments as follows.

Reviewer 2 Report
This paper presents a study on the usability of the use of smart glasses (Google Glass) for training nursing skills and interprofessional interactions in emergency situations.
The application of smart glasses in this context is interesting and relevant and studies about the adoption and usability of such systems are clearly important.
The paper however needs a lot of work, in my view. The main issues I find are: unclear methodology; confusing presentation of results; lack of analysis of prior work.
Regarding the methodology, there are many issues that are nor well explained:
- If the main purpose of the study was to evaluate the usability of the proposed system, why have a methodology based on an experiment? Why was a control group needed, and what was measured about this group? There are almost no results presented for the control group (except for the fist table of results with the characteristics of the participants).
- It is also not clear what procedure was followed. How were participants selected and assigned to each group? How were participants assigned to their roles? What was asked that participants do exactly? Was their performance measured? How? What was the end goal of the procedure that participants went through?
- There is no description of how the participants used the system. In the discussion, the paper states that ". participants attempted to use various features of the smart glasses to be actively engaged in team working and clinical decision... " but this is actually not described in the paper. What features did participants use? The features of the system are also not well described.
- There is confusion in the way that the number of participants is presented. In line 138, the paper states that 32 participants were recruited, but than Table 2, mentions 31 participants in the intervention plus 30 in the control...
Regarding the presentation of results, the section of results seems a dump of tables with little synthesis to help the reader. The results should be analysed and explored and better ways to convey them might be sought. For example, for some parts, graphics may be more suited. The instruments that were used should also be described in more detail (sections 2.5.1, 2.5.2, 2.5.3, 2.5.4). What were these instruments measuring, exactly?
(Section 2.6 is repeated from section 2.5)
Regarding the analysis of prior work, I was expecting to see a section on Related work where similar uses of smart glasses have been made. This would make it more obvious what the scientific contribution of this paper is in relation to other works.
Smaller issues:
The abstract mentions various averages but these are meaningless in the abstract because it does not say which scale was used.
There are various writing errors. The paper should be completely checked for correct spelling and writing mistakes.
Author Response

(The authors gave the same response as above.)

Reviewer 3 Report
First, congratulations on writing such a valuable article. I would also like to thank you for your dedication to writing the article.
However, I propose to improve the article. Detailed comments are included in the .pdf file.

Author Response

(The authors gave the same response as above.)

Reviewer 4 Report
The manuscript presents an interesting study of smart glasses utilization in nursing students training. While the study is interesting and the manuscript is nicely written, there are some issues, which, in my opinion, prevent it to be considered acceptable in the current form.
The most prominent issue is that the originality and novelty of the work is not established as the manuscript lacks a proper “Related Work” section. There are several recent works dealing with the utilization of smart glasses in nurse education, for example the works of Byrne, Senk, Frederick, Van Gelderen or Kopetz. The authors should expand the introduction section or add a separate section to compare their contribution to related works such as these. With respect to this, the authors may also take a look at a related review paper https://www.mdpi.com/2227-9032/9/8/947.
Some introductory sentences should be added to Section 2.5. It should explain why exactly the instruments, described in its subsections have been used. The subsections (2.5.1, ...) themselves should also be improved as now they look more as draft notes than a final text.
Provided that such data are available, it should be useful to add a comparison with the same or similar training performed without the use of the smart glasses.
There are also some minor issues that should be addressed:
- Some explanation of the term “high-fidelity simulator” is needed as Electronics is not a healthcare-centric journal.
- Please, use quotes instead of apostrophes on line (l.) 53.
- Some citations supporting the claim “the benefits of TBS have been well recognized” (l. 57) should be added.
- A reference or link to METIman (l.131) should be provided.
- A reference or link to SPSS (l.180) should be provided.
The manuscript contains several typos and fragments that need correction. Some of them are listed below:
- 7ability (line (l.) 48),
- focuses (l.49)
- little no previous experience of smart glass (l.94)
- used that patients (l.96)
- is as follow (l.128)
- contents (l.191)
- I would participate this kind of simulation again (Table 6)
- Majority participants (l.209)
- inexperienced devices (l.301)
- an effective means (l.331)
Author Response

(The authors gave the same response as above.)

Reviewer 5 Report
The context and the state of the art are well introduced, however, there are many other studies using virtual or augmented reality and remote collaborative assistance in various other fields for assistance, for example to maintenance operators, but quite few for the training phases. However, there are collaborative learning (training) systems, even in the health sector using virtual reality. The idea of using augmented reality and remote collaboration for training both the bedside team and the remote support team is therefore interesting.
The method followed as described in the paper is right. The smart-glass based system consists of four parts: lecture ; students developing algorithm for patient care ; team based training and evaluation. Participants had a lecture for analysis, medication and nursing care for patients.
Regarding the results, the quantitative answers are interesting for the questions that were asked. The responses are quite encouraging as they show overall user satisfaction.
About qualitative responses comments are objective and are good complements to the quantitative results, as confirming them. Even if the results are quite positive, the recommendations made for future improvements are quite relevant.
Author Response
We are grateful for the positive and constructive comments that originated for the review process.
We have revised the manuscript according to reviewers’ comments.

Round 2
Reviewer 2 Report
Authors have made a substantial effort to fix the issues found in the previous round. I have no further objections to publication.
Author Response
We sincerely appreciate your time to review the manuscript.
We have recheked the manuscript and made some minor revisions.
Reviewer 3 Report
There are no comments.
Author Response

(The authors gave the same response as above.)

Reviewer 4 Report
I am glad to conclude that the revised version addressed the issues of the original manuscript in a satisfactory way.
There are still ways in which the manuscript can be improved, some suggestions are listed below:
- Line (L.) 22: The square brackets should be avoided here as the numbers inside can be mistaken for citations.
- Please, add a new line with information about your study to Table 1.
- L. 441: Change “This study had a few limitations” to “This study had few limitations”.
Author Response
We sincerely appreciate your time to review the manuscript.
We have recheked the manuscript and made some minor revisions as follows.
Line (L.) 22: The square brackets should be avoided here as the numbers inside can be mistaken for citations.
-> The squre brackets were removed and the sentence was revised.
L. 441: Change “This study had a few limitations” to “This study had few limitations”.
-> The sentence was revised reflecting reviewer's comments